# Novel *EVI5*::*BRAF* Gene Fusion in Infantile Fibrosarcoma: A Case Report and Review of Literature

**DOI:** 10.3390/ijms26031182

**Published:** 2025-01-29

**Authors:** Judith González-López, Luis Alberto Rubio-Martínez, Miryam Atarés, José Vicente Amaya, Maria Carmen Huart, Empar Mayordomo-Aranda, Francisco Giner

**Affiliations:** 1Pathology Department, Hospital Universitari i Politècnic La Fe, 46026 València, Spain; judithg2993@gmail.com (J.G.-L.); rubio_lui@gva.es (L.A.R.-M.); huart_mca@gva.es (M.C.H.); mayordomo_emp@gva.es (E.M.-A.); 2Radiology Department, Hospital Universitari i Politècnic La Fe, 46026 València, Spain; miryamrx@gmail.com; 3Orthopaedics and Traumatology Department, Hospital Universitari i Politècnic La Fe, 46026 València, Spain; amaya_jos@gva.es; 4Pathology Department, Universitat de València, Avinguda Blasco Ibáñez, 15, 46010 València, Spain

**Keywords:** infantile intraosseous fibrosarcoma, *BRAF* translocation, *EVI5::BRAF* fusion

## Abstract

Infantile Fibrosarcoma is a malignant tumor of fibroblastic origin, typically found in early childhood, locally aggressive, and characterized by molecular alterations that activate tyrosine kinase signaling, primarily the *ETV6::NTRK3* fusion. In recent years, a series of fusions different from the classic one have been described, including *NTRK1*, *RAF1*, and *BRAF*. In this paper, we present a case of IFS with a novel *EVI5::BRAF* fusion. We observed a spindle cell neoplasm growing in intertwined fascicles within a fibrous stroma, without the formation of an immature osteoid matrix. Weak and focal immunoreactivity for S100 was observed. SATB2 exhibited diffuse and intense staining, with focal expression of osteonectin and negativity for caldesmon, Smooth Muscle Actin, desmin, GFAP, SOX10, MelanA, panTRK, and HMB45. The Ki67 index was 7%, and the tumor harbored an *EVI5::BRAF* genetic fusion. To the best of our knowledge, the *EVI5::BRAF* fusion has not yet been described in *BRAF* fusions in IFS. Nevertheless, further studies are needed to define the prognostic features of these emerging *BRAF* sarcomas, along with new anti-*BRAF* therapeutic approaches.

## 1. Introduction

Pediatric mesenchymal tumors encompass a heterogeneous group of benign and malignant pathologies, both de novo (90%) and associated with hereditary syndromes (10%). Soft tissue sarcomas are more common in children than in adults, with an incidence that increases with age from 0.9 cases per 100,000 person-years in children under 10 years old to 18.2 cases per 100,000 person-years in older children. In children under 15 years old, the most common sarcoma is rhabdomyosarcoma (95–97% of cases), while the remaining 3–5% includes both low-grade tumors (rarely metastatic or locally aggressive) and high-grade tumors [1].

This group includes both typical pediatric sarcomas and other “adult-type” sarcomas that can appear in childhood. One of the typical pediatric tumors, Infantile Fibrosarcoma (IFS), is a malignant tumor of fibroblastic origin, typically found in early childhood, locally very aggressive, and characterized by molecular alterations that activate tyrosine kinase signaling, primarily the *ETV6*::*NTRK3* fusion. Despite its local aggressiveness, metastases are infrequent (8–15%), and the 10-year survival rate is 90% with surgical and chemotherapeutic treatment. Local recurrence occurs in 25–40% of cases with affected margins. These tumors exhibit a poorly characteristic immunohistochemical profile, with variable expression of CD34, S100, Smooth Muscle Actin, and desmin. PanTRK may be positive in cases where mutations are present [1].

In recent years, a series of IFSs with fusions different from the classic one have been described, involving other tyrosine kinase genes, including *NTRK1*, *NTRK2*, *RET*, *MET*, and *RAF1* [1]. Among these new fusions is *BRAF*, a gene that encodes proteins from the RAF family of protein kinases that regulate the MAP kinase/ERK signaling pathway. This pathway promotes cell proliferation and survival, and its activation induces tumorigenesis [2]. Although *BRAF* has functions typically associated with carcinomas, its role as an inducer of mesenchymal neoplasms has also been described. These neoplasms characteristically present a “spindle cell” or “epithelioid” pattern and are *NTRK*-negative [2]. This group includes not only IFSs but also mandibular [3] and central nervous system sarcomas [4,5], as well as myxoinflammatory myofibroblastic sarcomas, glomus tumors, stromal tumors of the kidney, and gastrointestinal stromal tumors (GISTs) [6].

Treatment consists of surgery and chemotherapy. Despite being locally aggressive, metastasis is rare (10%), and the 10-year survival rate exceeds 90%. The most relevant factor for recurrences (which occur in 25–40% of cases) is resection with clear margins, which is why surgery is often aggressive. Given the development of new therapeutic targets against *NTRK* and *BRAF*, the identification of these molecular alterations is of vital importance for the prognosis and treatment of patients. [1].

In this paper, we present a case of IFS with a novel *EVI5*::*BRAF* fusion and discuss its clinical, histological, immunohistochemical, and molecular features.

## 2. Case Presentation

### 2.1. Clinical and Radiological Findings

A 14-year-old male experienced intense pain in his left arm after a sudden movement while descending a slide at a water park. He sought emergency care and was diagnosed with a pathological fracture of the left humerus. The patient denied any prior symptoms before the accident. An arm sling was applied by orthopedics, and 20 days later, he returned for a follow-up magnetic resonance imaging (MRI) scan.

The MRI revealed a pathological fracture of the proximal humerus with angulation, located over a poorly defined lytic lesion in the proximal diaphysis, measuring approximately 17 × 72 mm. There was a cystic area of 13 mm in its proximal portion, cortical erosion, and intense contrast uptake. The findings suggested a differential diagnosis of fibrous dysplasia (FD), an aneurysmal bone cyst (ABC), Ewing sarcoma (ES), and osteosarcoma (OS). Additionally, there was edema and contrast uptake in the adjacent musculature, which could be secondary to inflammatory changes due to the fracture, although infiltration could not be ruled out (Figure 1). Given these findings, the patient was referred to our center and was admitted for further evaluation.

A core needle biopsy (CNB) was performed. Three cores were obtained, with a diagnosis of “low-grade sarcoma”. A thoracoabdominal–pelvic computed tomography (CT) scan was performed, showing no signs of distant metastasis. After discussion in the sarcoma committee, a decision was made to proceed with radical surgery, including resection and reconstruction with a vascularized fibula graft. Medical Oncology determined that there was no indication for Neoadjuvant Chemotherapy (NCT) or Radiotherapy (NCRT).

During surgery, 13 cm of the humerus was resected along with the coracobrachialis muscle and parts of the deltoid and brachialis anterior muscles. Intraoperative samples were sent to the Pathology Department, confirming free margins. The humerus was reconstructed using a free osteocutaneous fibula flap from the right leg (intramedullary in the proximal humerus and trench technique in the distal humerus). After the surgery, the patient was discharged without complications, with no indication for adjuvant therapy from Medical Oncology.

### 2.2. Macroscopic Findings

The resection specimen of the humerus measured 14 × 5.4 × 4.6 cm. After staining the margins and sectioning the specimen, a poorly defined yellowish lesion measuring 4.7 × 2.3 cm was observed within the medullary cavity, expanding into the surrounding soft tissues. The lesion was located 2.6 cm from the proximal margin, 1.7 cm from the distal margin, and 1 cm from the circumferential margin, all of which were macroscopically free of the lesion (Figure 2).

### 2.3. Microscopic Findings

The lesion exhibited a “fish spine” pattern in the medullary cavity, composed of atypical cells with moderate pleomorphism and medium-to-large size. These cells were arranged in long, highly intertwined fascicles, breaking through the cortical bone and extending into the soft tissues (Figure 3a,b). The nuclei were oval to elongated, hyperchromatic, and granular with nuclear pseudo-inclusions, and the cytoplasm was elongated and eosinophilic (Figure 3c). An inflammatory component, predominantly mononuclear and plasmacytic, was observed between the tumor fascicles and around the blood vessels. There were 5 mitoses in 10 high-power fields (Figure 3d). No bone trabeculae, osteoid formation, or areas of necrosis were present within the lesion.

Based on the histological findings and the results of the previous biopsy, the diagnosis was oriented towards IFS. However, other diagnoses based on age, appearance and location such as low-grade osteosarcoma (LGO), inflammatory myofibroblastic tumor (IMT), malignant peripheral nerve sheath tumor (MPNST), or melanoma were also considered. Taking into account these differential diagnoses, microscopic findings and immunohistochemical stainings were crucial, in addition to Next-Generation Sequencing (NGS).

### 2.4. Immunohistochemical Findings

Weak and focal immunoreactivity was observed for S100. SATB2 expressed a diffuse and intense staining, with focal expression of osteonectin and negativity for caldesmon, Smooth Muscle Actin (SMA), desmin, GFAP, SOX10, MelanA, panTRK, and HMB45. H3K27me3 was not valuable, and the Ki67 index was 7% (Figure 4 and Table 1). According to the Fédération Nationale des Centres de Lutte Contre le Cancer (FNLCC) criteria [7], the score was 4, corresponding to a G2 grade (Dedifferentiation 3/3, mitosis 1/3, and necrosis 0/2). Table 1 includes the antibodies used for diagnosis.

### 2.5. Fluorescence In Situ Hybridization (FISH) and Next-Generation Sequencing (NGS) Results

Given the suspicion of pediatric fibrosarcoma and the negativity for panTRK, fluorescence in situ hybridization (FISH) was performed for the *BRAF* gene, revealing an *EVI5::BRAF* fusion. This rearrangement was later studied by Next-Generation Sequencing (Figure 5), confirming the fusion.

It is important to note that although the tumor expressed both SATB2 and osteonectin, a diagnosis of LGS was ruled out due to the cytomorphological features, lack of osteoid matrix and absence of *MDM2* amplification. All these findings supported the IFS diagnosis considering SATB2 and osteonectin expression as an unspecific staining. Additionally, a diagnosis of IMT was ruled out due to the aggressive local behavior of the lesion and the absence of ALK immunexpression and translocation. IFS can also present with a prominent lymphoid inflammatory component [1].

Table 2 includes a clinical and pathological summary of the case.

## 3. Discussion

Among the *BRAF* fusions described in IFS are *SEPT7*::*BRAF*, *SEPT9*::*BRAF*, *SEPT11*::*BRAF*, *ERC1*::*BRAF, PDE10A*::*BRAF, CUX1*::*BRAF* and *KIAA1549*::*BRAF* [2,7,8,9]. Kao et al. [9], in his series of Infantile Fibrosarcomas, describes five cases with *BRAF* fusions in patients of varying ages (from 2 days to 16 years), with three tumors located in the pelvis, one in the T6 vertebra, and another in the retroperitoneum. The tumors exhibited a ‘spindle cell’ morphology with few mitoses and patchy and focal expression for SMA, with an absence of expression for desmin and S100. It is important to note that these tumors are not associated with the upregulation of *BRAF* mRNA, so the immunohistochemistry of BRAF is not reliable.

*EVI5* (Ecotropic Viral Integration Site 5) is a gene that enables GTPase activator activity and small GTPase binding activity, and it is involved in the positive regulation of GTPase activity and retrograde transport from the endosome to the Golgi. It functions as a regulator of cell cycle progression by stabilizing the FBXO5 protein and promoting cyclin-A accumulation during interphase [10]. *EVI5* was discovered by X. Liao et al. [11] in 1995 and defined as a common site of retroviral integration in T-cell lymphomas in mice. Since then, it has been described as a regulatory gene involved in both proliferation and metastasis in tumors such as non-small-cell lung carcinoma [12] and hepatocellular carcinoma [13], as well as in neuroblastoma, where chromosomal translocation may influence its origin. [14]. To the best of our knowledge, the *EVI5*::*BRAF* fusion has been only described in congenital melanocytic nevi [15].

Kao et al. [9] describes that these “IFS-like” tumors appear in unusual clinical settings, such as older age groups or intra-abdominal locations, unlike classic IFS, which is more common in the early years of life and in the extremities. It is important to consider that the radiological appearance of the tumor is highly variable, potentially presenting as ovoid, nodular, or fascicular, with irregular contrast uptake and poorly defined margins. In this context, core needle biopsy is crucial and enables a definitive diagnosis to guide therapeutic decision-making [16]. The Penning series (2021) includes 14 tumors with *BRAF* fusions, making it the largest series reported to date. Similar to Kao, these tumors are defined as “IFS-like” [6].

These tumors have similar prognoses, making it complex to determine whether they correspond to variants of IFS or distinct entities. However, in the latest classification of pediatric tumors by the WHO (2022), they are considered as IFS, and *BRAF* fusions are included alongside the classic *NTRK* ones [1]. These genetic alterations are also described as oncogenic drivers [6].

It is important to highlight that while tumors harboring the *BRAF V600E* mutation have shown a good response to anti-*BRAF* therapy, those with other mutations or fusions have demonstrated resistance or paradoxical activation. Consequently, new clinical trials are currently under development, focusing primarily on *MEK/BRAF* inhibitors [6]. In recent years, theranostics has emerged as a novel form of precision medicine that integrates diagnostics with therapeutics, aiming to develop more personalized alternatives for each patient through the use of nanoparticles. In both the present and the future, these and other innovative therapeutic alternatives can be applied even in difficult and singular cases, such as the one proposed, extending beyond immunology and genetics [17].

To the best of our knowledge, *EVI5*::*BRAF* has not been described in *BRAF* fusions in IFS yet, making this case the first in the literature. Nevertheless, further studies are needed to define the prognostic features of these emergent *BRAF* sarcomas. This will not only help to guide diagnosis in cases where the same fusion is detected, but it will also serve as a foundation for the development of new therapeutic targets in the future.

## Figures and Tables

**Figure 1 ijms-26-01182-f001:**
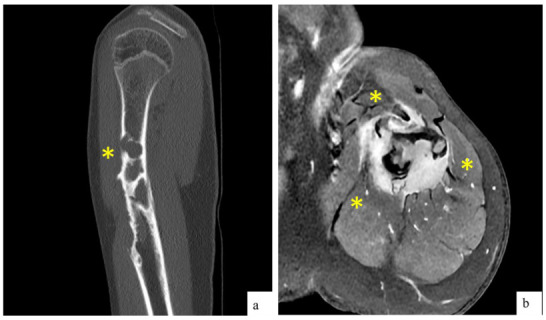
(**a**) CT findings: Sagital reconstruction showing diaphyseal lytic lesion with cortical erosion, causing bone remodeling. (**b**) MRI findings: Post-contrast transverse T1 sequence showing surrounding soft tissue mass with homogeneous and intense contrast uptake. In both images, the lesion is delineated between the yellow asterisks.

**Figure 2 ijms-26-01182-f002:**
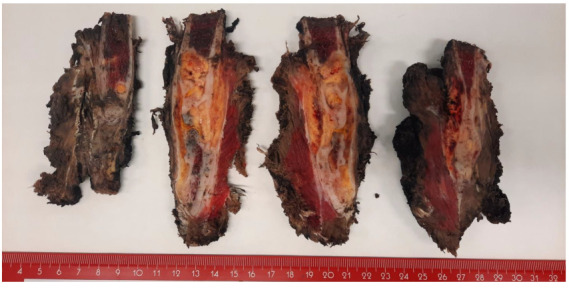
Macroscopic findings. A poorly defined yellowish lesion measuring 4.7 × 2.3 cm was observed within the medullary cavity, expanding into the surrounding soft tissues.

**Figure 3 ijms-26-01182-f003:**
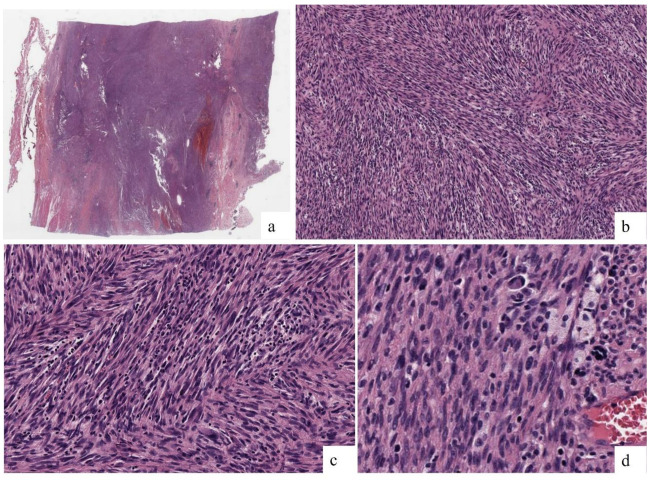
Microscopy findings. (**a**) The lesion occupied the medullary cavity, breaking through the cortex and extending into the surrounding soft tissues (HE; 4×). (**b**) Diffuse ‘herringbone’ pattern (HE; 10×). (**c**) Elongated and interwoven fascicles composed of atypical cells with moderate pleomorphism and medium size, with hyperchromatic nuclei and poorly defined cytoplasm. (HE; 20×). (**d**) Isolated mitoses were observed, with a notable lymphoid inflammatory component. No osteoid formation and areas of necrosis were present within the lesion. (HE, 40×).

**Figure 4 ijms-26-01182-f004:**
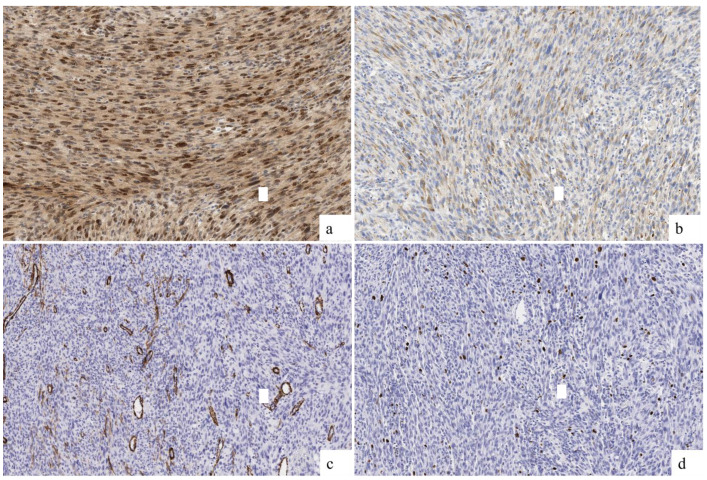
Immunohistochemical findings. (**a**) Diffuse positivity for SATB2 (20×). (**b**) Weak and focal positivity for osteonectin (20×). (**c**) Absence of expression for Smooth Muscle Actin (20×). (**d**) Ki67 index was established in 7% (20×).

**Figure 5 ijms-26-01182-f005:**
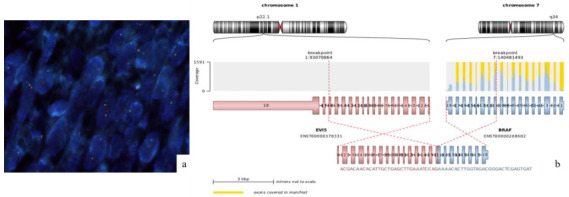
FISH and NGS results. (**a**) FISH shows *BRAF* fusion. (**b**) NGS shows the fusion point between *EVI5* and *BRAF*. Made with the SeqOne^®^ in vitro diagnostic medical device.

**Table 1 ijms-26-01182-t001:** Main antibodies and staining pattern used.

Antibody	Source	Clone	Staining Pattern
S100	Dako	Polyclonal	Diffuse (Nuclear)
SATB2	Zeta	ZR167	Diffuse (Nuclear)
Caldesmon	Dako	h_CD	Negative
Smooth Muscle Actin	Dako	HHF-35	Negative
Desmin	Dako	D33	Negative
Osteonectin	Leica	G-15-G12	Focal (Cytoplasmic)
GFAP	Dako	Polyclonal	Negative
SOX10	Biocare	BC34	Negative
MelanA	Dako	A103	Negative
HMB45	Dako	HMB-45	Negative
H3K27me3	Gennova/Biocare	C36B11	Not evaluable
Ki67	Dako	MIB-1	7%
ALK	Dako	ALK1	Negative

**Table 2 ijms-26-01182-t002:** Clinical and pathological summary. *SMA: Smooth Muscle Actin. Mt: metastases*.

Age	Sex	Site	S100	SMA	SATB2	Ki67	Mt	Genetics	Treatment	Outcome
14	Male	Left humerus	Weak, focal	Negative	Intense, difusse	7%	No	EVI5::BRAF	Resection and reconstruction	No evidence of disease

## Data Availability

Data is unavailable due to privacy or ethical restrictions.

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
