# Peer review of "Novel EVI5::BRAF Gene Fusion in Infantile Fibrosarcoma: A Case Report and Review of Literature"

_ijms, 2025, doi:10.3390/ijms26031182_

Round 1
Reviewer 1 Report
Comments and Suggestions for Authors
The title “Novel EVI5::BRAF Gene Fusion in Infantile Fibrosarcoma: A Case Report and Review of literature” is appropriate for the content covered.
In this manuscript, 61 clinical, histological, immunohistochemical, and molecular features are examined using radiological, macroscopic, microscopic, and immunohistochemical techniques in order to further understand the case of infantile fibrosarcoma with a novel EVI5::BRAF fusion. However, the manuscript has some shortcomings in the area of image analysis and validation of the performance of the proposed method.
The topic is very interesting and current, but the manuscript setting needs to be improved. To make the document more validated, some sections need to be substantially expanded, particularly following the comments.
Please see the comments below.
1) Authors are strongly encouraged to consider the following state of the art:
a) Since the paper is not very innovative in terms of the techniques used, it would be appropriate for the authors to expand the pathological framework at the level of the introduction or discussion and conclusion in which the theranostic field (for diagnosis and therapy) is mentioned as a new approach to personalized medicine even in difficult and singular cases such as the one proposed. The authors are strongly advised to consider [10.3390/cancers16193323] potential methods for enhancing difficult-to-manage cancer cases (such as gastric cancer) prognosis and treatment in the future. This would make this case report much more impactful in the scientific community and not limited to basic genomics and immunohistochemistry.
b) [10.1038/s41379-021-00806-w] authors are also advised to follow this manuscript, particularly its structure. In particular, authors are advised to insert a table with clinicopathological characteristics and genetics.
These suggestions will be essential and are recommended to the authors so that the limitations and strengths of the study are widely highlighted; furthermore this evaluation of the validity of the methods, results and interpretation of the data will bring a higher scientific impact of this promising work.
2) A list of abbreviations at the end of the manuscript would help make the manuscript more explanatory. Authors are requested to include them.
3) Insert arrows pointing to the parts shown in Figure 1.
4) Authors are strongly recommended to include a graphical or tabular summary of the statistical analysis performed, clarifying which methods were adopted for reproducibility and accuracy and for performance evaluation of the proposed applied method.
5) English is quite understandable and does not require any particular improvement.
Finally, it would be helpful to extend the references to enhance the coherence of the article.
Author Response
The title “Novel EVI5::BRAF Gene Fusion in Infantile Fibrosarcoma: A Case Report and Review of literature” is appropriate for the content covered.
In this manuscript, 61 clinical, histological, immunohistochemical, and molecular features are examined using radiological, macroscopic, microscopic, and immunohistochemical techniques in order to further understand the case of infantile fibrosarcoma with a novel EVI5::BRAF fusion. However, the manuscript has some shortcomings in the area of image analysis and validation of the performance of the proposed method.
The topic is very interesting and current, but the manuscript setting needs to be improved. To make the document more validated, some sections need to be substantially expanded, particularly following the comments.
Please see the comments below.
1) Authors are strongly encouraged to consider the following state of the art:
- a) Since the paper is not very innovative in terms of the techniques used, it would be appropriate for the authors to expand the pathological framework at the level of the introduction or discussion and conclusion in which the theranostic field (for diagnosis and therapy) is mentioned as a new approach to personalized medicine even in difficult and singular cases such as the one proposed. The authors are strongly advised to consider [10.3390/cancers16193323] potential methods for enhancing difficult-to-manage cancer cases (such as gastric cancer) prognosis and treatment in the future. This would make this case report much more impactful in the scientific community and not limited to basic genomics and immunohistochemistry.
Reply 1a: Thank you for your revision and pointing this out. We agree, therefore, we have added a new paragraph with insights on the theranostic field (Page 10, line 259).
- b) [10.1038/s41379-021-00806-w] authors are also advised to follow this manuscript, particularly its structure. In particular, authors are advised to insert a table with clinicopathological characteristics and genetics.
Reply 1b: Agree. We have reviewed the article and added a new table with clinicopathological information (Table 2) and new insights (Page 3, line 80 / Page 9, line 254 / Page 10, line 259).
2) A list of abbreviations at the end of the manuscript would help make the manuscript more explanatory. Authors are requested to include them.
Reply 2: We added a nre list of abbreviations at the end of the manuscript. Page 10, line 269.
3) Insert arrows pointing to the parts shown in Figure 1.
Reply 3: Thank you for pointing this out. We have added asterisks in Figure 1 to delineate the lesión and make it easier to ubicate.
4) Authors are strongly recommended to include a graphical or tabular summary of the statistical analysis performed, clarifying which methods were adopted for reproducibility and accuracy and for performance evaluation of the proposed applied method.
Reply 4: Since this is a report of a single case, we have not performed statistical analyses for this article.
5) English is quite understandable and does not require any particular improvement.
Reply 5: Thank you!
Finally, it would be helpful to extend the references to enhance the coherence of the article.
Reply 6: We have added new references, bringing the total to 18.
Reviewer 2 Report
Comments and Suggestions for Authors
Interesting case report. However, I have some minor comments:
- For a case report, the methods section in the abstract should be removed.
- Please state about the novelty of your case report, both in the Introduction and in the end of your manuscript. And additionally please mention the clinical value of your findings.
- Please mention that fibrosarcomas can appear in any shape on MRI making the diagnosis difficult. Please use the following paper for that finding: https://doi.org/10.1007/s00238-020-01669-1
Author Response
Interesting case report. However, I have some minor comments:
- For a case report, the methods section in the abstract should be removed.
Reply 1: We have removed the "Methods" section from the abstract as per your suggestions.
- Please state about the novelty of your case report, both in the Introduction and in the end of your manuscript. And additionally please mention the clinical value of your findings.
Reply 2: Thank you for your comment. We have emphasized the novelty of the fusion (See page 3, line 87 / Page 10, line 264) and its therapeutic implications (Page 9, line 255)
- Please mention that fibrosarcomas can appear in any shape on MRI making the diagnosis difficult. Please use the following paper for that finding: https://doi.org/10.1007/s00238-020-01669-1
Reply 3: Thank you for your suggestion. We agree and have added this information (Page 9, line 244).
Reviewer 3 Report
Comments and Suggestions for Authors
This study presents a case of infantile fibrosarcoma (IFS) featuring a novel EVI5::BRAF fusion, which has not previously been reported in the context of BRAF fusions in IFS. As such, this case represents the first documentation of this specific fusion in the literature.
The manuscript is well-organized and clearly presented, offering thorough and compelling evidence to support the findings. The information provided is both comprehensive and sufficient, effectively contributing to the understanding of this rare genetic fusion.
Author Response
This study presents a case of infantile fibrosarcoma (IFS) featuring a novel EVI5::BRAF fusion, which has not previously been reported in the context of BRAF fusions in IFS. As such, this case represents the first documentation of this specific fusion in the literature.
The manuscript is well-organized and clearly presented, offering thorough and compelling evidence to support the findings. The information provided is both comprehensive and sufficient, effectively contributing to the understanding of this rare genetic fusion.
Reply 1: Thank you for your words! They are highly appreciated.
Round 2
Reviewer 1 Report
Comments and Suggestions for Authors
The authors improved the manuscript quality by positively resolving comments. They also responded comprehensively to the entire review, giving consistency to the work presented.